# A Peptide-Based HIV-1 Fusion Inhibitor with Two Tail-Anchors and Palmitic Acid Exhibits Substantially Improved In Vitro and Ex Vivo Anti-HIV-1 Activity and Prolonged In Vivo Half-Life

**DOI:** 10.3390/molecules24061134

**Published:** 2019-03-21

**Authors:** Shan Su, Giselle Rasquinha, Lanying Du, Qian Wang, Wei Xu, Weihua Li, Lu Lu, Shibo Jiang

**Affiliations:** 1Key Laboratory of Medical Molecular Virology of MOE/MOH, School of Basic Medical Sciences and Shanghai Public Health Clinical Center, Fudan University, 130 Dong An Rd., Xuhui District, Shanghai 200032, China; 14111010010@fudan.edu.cn (S.S.); wang_qian@fudan.edu.cn (Q.W.); xuwei0576@126.com (W.X.); 2Lindsley F. Kimball Research Institute, New York Blood Center, New York, NY 10065, USA; glr@outlook.com (G.R.); ldu@nybc.org (L.D.); 3NHC Key Laboratory of Reproduction Regulation, Shanghai Institute of Planned Parenthood Research, Fudan University, Shanghai 200032, China; iamliweihua@foxmail.com

**Keywords:** HIV-1, gp41, fusion inhibitor, six-helix bundle, peptide, palmitic acid

## Abstract

Enfuvirtide (T20) is the first U.S. FDA-approved HIV fusion inhibitor-based anti-HIV drug. Its clinical application is limited because of its low potency and short half-life. We previously reported that peptide HP23-E6-IDL, containing both N- and C-terminal anchor-tails, exhibited stronger potency and a better resistance profile than T20. Here we designed an analogous peptide, YIK, by introducing a mutation, T639I, and then a lipopeptide, YIK-C16, by adding palmitic acid (C16) at the C-terminus of YIK. We found that YIK-C16 was 4.4- and 3.6-fold more potent than HP23-E6-IDL and YIK against HIV-1_IIIB_ infection and 13.3- and 10.5-fold more effective than HP23-E6-IDL and YIK against HIV-1_Bal_ infection, respectively. Consistently, the ex vivo anti-HIV-1_IIIB_ activity, as determined by the highest dilution-fold of the serum causing 50% inhibition of HIV-1 infection, of YIK-C16 in the sera of pretreated mice was remarkably higher than that of YIK or HP23-E6-IDL. The serum half-life (t_1/2_ = 5.9 h) of YIK-C16 was also significantly longer than that of YIK (t_1/2_ = 1.3 h) and HP23-E6-IDL (t_1/2_ = 1.0 h). These results suggest that the lipopeptide YIK-C16 shows promise for further development as a new anti-HIV drug with improved anti-HIV-1 activity and a prolonged half-life.

## 1. Introduction

Because of the widespread application of highly active antiretroviral therapy (HAART), AIDS-related deaths have decreased by more than 51% since the peak in 2004 (http://www.unaids.org/en/resources/fact-sheet). The essence of HAART is the combination of various drugs that act on different stages of HIV-1 infection and replication [1]. So far, the U.S. FDA-approved HIV drugs mainly consist of four categories, including nucleoside/nonnucleoside reverse transcriptase inhibitors (NRTIs/NNRTIs), protease inhibitors (PIs), integrase inhibitors (IIs), and entry inhibitors (EIs) (https://www.fda.gov). Among them, NRTIs/NNRTIs, PIs, and IIs take effect by blocking the replication process after HIV enters the target cell, while EIs have the advantage of working at the early stage of HIV entry, thus preventing viral fusion into the host cell [2].

The entry process of HIV is triggered by the binding between the gp120 subunit of the HIV envelope glycoprotein (Env) and the host receptor CD4 and co-receptor, CCR5 or CXCR4, on the target cell. Then, the other Env subunit, gp41, is exposed, and the N-terminal fusion peptide (FP) of gp41 inserts into the target cell membrane. The gp41 N-terminal heptad repeat (NHR) domain then interacts with the homologous C-terminal heptad repeat (CHR) domain to form a hairpin-like 6-helix bundle (6-HB) structure, consisting of a NHR trimer core with three CHR helices packing into the hydrophobic grooves on the surface of the NHR trimer in an antiparallel way [3,4,5]. 6-HB pulls the viral particle and host cell into proximity and elicits membrane fusion between virus and cells. Fusion inhibitors, a subtype of EIs, are derived from the NHR [6] or CHR [7] domain of gp41, and inhibit the fusion between HIV and target cell by competitively blocking the formation of the homologous 6-HB structure. However, clinical application of the only U.S. FDA-approved HIV fusion inhibitor, enfuvirtide (T20), has been limited by its low potency and susceptibility to drug resistance [8]. Moreover, T20 requires twice-daily injection in clinical use because of its short in vivo half-life [9].

Recently, we discovered that the addition of three residues, Ile654, Asp655, and Leu656 (IDL), which adopt a hook-like (tail-anchor) structure, to the C-terminus of CHR peptides can greatly enhance their binding to NHR and anti-HIV-1 activities [10], providing a new strategy to design and optimize HIV-1 fusion inhibitors. On the basis of the IDL hook structure, we designed a 32-mer peptide inhibitor (HP23-E6-IDL), which exhibited potent inhibitory activity against HIV-1 of diverse subtypes and tropisms, especially those resistant to other HIV-1 fusion inhibitors [11]. However, similar to other peptide-based drugs, HP23-E6-IDL possesses a short in vivo half-life, which significantly attenuates its potential for development into a new anti-HIV drug. Emerging studies have proven that the conjugation of a lipid domain to a peptide substantially enhances its antiviral activity and in vivo stability [12,13,14], possibly resulting from the preferential interaction of the lipid domain with the lipid rafts of membranes where HIV-1 fusion occurs, thereby increasing the topical concentration of the peptide, prompting our exploitation of lipid conjugation to further enhance the inhibitory activity of HP23-E6-IDL.

## 2. Results

### 2.1. YIK-C16 was Highly Potent in Inhibiting HIV-1 Infection In Vitro

We modified HP23-E6-IDL with two strategies. First, mutation T639I, which was reported to enhance the potency of fusion inhibitors [15], was introduced to HP23-E6-IDL to generate an analog peptide, YIK. Second, palmitic acid (C16) was conjugated to the C-terminus of YIK with a linker (GSG-PEG4-K) between C16 and YIK (Figure 1a). As shown in Figure 1b,c, the inhibitory activity of YIK against HIV-1 infection was only slightly better than that of HP23-E6-IDL. However, the resultant lipopeptide YIK-C16 possessed dramatically increased anti-HIV-1 activity in vitro. Its half maximum inhibitory concentrations (IC_50_) for inhibiting HIV-1_IIIB_ and HIV-1_Bal_ infection are 76 and 61 pM, respectively, which are about 4.4- and 13.3-fold more potent than HP23-E6-IDL. Our previous study showed that HP23-E6-IDL could inhibit infection by T20- and HP23-resistant variants. Here, we also tested the inhibitory activities of YIK and YIK-C16 to HIV-1 mutants resistant to T20, HP23, and T2635 (a 38-mer fusion inhibitor). As shown in Table 1, YIK exhibited inhibitory activity similar to that of HP23-E6-IDL against both the wild-type (WT) and resistant mutants with IC_50_s ranging from 0.5 to 5.2 nM, while YIK-C16 showed much higher potency against these resistant mutants with IC_50_s ranging from 40 to 188 pM. YIK-C16 was about 12-, 12- and 26-fold more potent than HP23-E6-IDL for inhibiting infection by HIV-1 T20-, T2635-, and HP23-resistant strains, respectively. These results suggest that lipopeptide YIK-C16 possesses significantly enhanced inhibitory activity against in vitro infection of HIV-1 strains, including those resistant to the HIV-1 fusion inhibitors used in clinics or in developmental stages.

### 2.2. Secondary Structure of Complex Formed by YIK-C16 and an NHR Peptide

Circular dichroism spectrum (CD) was employed to investigate the secondary structure of the peptides HP23-E6-IDL, YIK, and YIK-C16, either alone, or in complex with NHR peptide N46 overlapping the sequence of the HIV-1 gp41 (residues 536-581). As shown in Figure 2a, the helical content of the HP23-E6-IDL/N46 mixture was much higher than that of HP23-E6-IDL, or N46 alone, indicating that HP23-E6-IDL can interact with N46 to form 6-HB. As for peptide YIK, the replacement of Thr639 by Ile did not change the secondary structure of HP23-E6-IDL, either alone or in complex with N46 (Figure 2b). Similarly, YIK-C16 could also form 6-HB with N46, and the helical content was also similar to that of HP23-E6-IDL and N46 (Figure 2c), suggesting that the addition of a lipid domain to the C-terminus of YIK does not significantly affect its α-helicity and its ability to interact with an NHR peptide to form 6-HB.

### 2.3. Improved Anti-HIV-1 Activity of YIK-C16 may Result from its Binding to the Cell Membrane

To investigate why YIK-C16 exhibited much higher potency than YIK and HP23-E6-IDL, we compared their ability to inhibit HIV-1-mediated cell-cell fusion. As shown in Figure 3a, YIK-C16 is about two-fold more potent than YIK and HP23-E6-IDL in inhibiting HIV-1-mediated cell-cell fusion, suggesting that the enhanced anti-HIV-1 activity of YIK-C16 may result from its improved ability to inhibit HIV-1 fusion with and entry into the host cell.

Most CHR-derived peptides inhibit HIV-1 infection and cell-cell fusion by binding to the viral gp41 NHR domain and blocking fusion-active 6-HB formation. Thus, we next tested whether YIK-C16 could inhibit 6-HB formation more efficiently. As shown in Figure 3b, YIK-C16 possessed inhibitory activity against 6-HB formation with as much potency as either HP23-E6-IDL or YIK, suggesting that conjugation of C16 to YIK does not affect the peptide’s ability to block 6-HB formation. Previous studies have shown that the addition of lipid domain to a CHR-peptide can enhance its binding to the cell membrane [16,17], indicating that the improved anti-HIV-1 potency of the lipopeptide may result from its enrichment on the cell surface where HIV-1 fusion occurs [18]. Therefore, we then tested whether lipopeptide YIK-C16 could attach to the cell membrane to inhibit HIV-1 infection. To make this determination, we preincubated MT-2 target cells with HP23-E6-IDL, YIK, or YIK-C16 at 4 °C for 30 min. The pretreated cells were washed to remove the unattached peptides, or remained unwashed to serve as a control, and were then infected by HIV-1_IIIB_. As shown in Figure 3c and 3d, without washing cells, both HP23-E6-IDL and YIK could inhibit HIV-1_IIIB_ infection with IC_50_ of 2.4 and 2.8 nM, respectively. However, after washing cells, HP23-E6-IDL and YIK could no longer inhibit HIV-1_IIIB_ infection, indicating that they do not bind to target cells before addition of HIV-1. In contrast, YIK-C16 could inhibit HIV-1_IIIB_ infection with IC_50_ around 930 and 90 pM when the pretreated target cells were washed and not washed, respectively (Figure 3e), suggesting that YIK-C16 can attach to and enrich the surface of cell membrane, thereby inhibiting HIV-1 infection with improved potency.

We then tested whether YIK-C16 could also bind to viral membrane to inhibit HIV-1 infection. HIV-1_IIIB_ viral particles were incubated with peptides at 4 °C for 30 min and then washed to remove the unbound peptides, or remained unwashed to serve as a control, and were then applied to infect MT-2 cells. As shown in Figure 3f,g, HP23-E6-IDL and YIK could inhibit HIV-1_IIIB_ infection without wash-away of peptides, whereas after washing, neither HP23-E6-IDL nor YIK could inhibit HIV-1_IIIB_ infection. In contrast, YIK-C16 could inhibit HIV-1_IIIB_ infection when the unbound peptide was washed away, suggesting that YIK-C16 could also bind to the viral membrane to enhance its antiviral activity.

### 2.4. YIK-C16 Exhibited Improved Ex Vivo Anti-HIV-1 Activity and Prolonged Serum Half-Life

Previous studies showed that the addition of a lipid domain could enhance in vitro and ex vivo anti-HIV-1 activity and improve the in vivo stability of peptide-based HIV-1 fusion inhibitors [12,13,14,19]. Here, we investigated whether YIK-C16 could also enhance ex vivo anti-HIV-1 activity, as determined by the highest dilution-fold of the serum causing 50% inhibition of HIV-1 infection (equivalent to the IC_50_) of the peptide and by extended in vivo half-life in mice. Twelve mice were randomly assigned into three groups and intraperitoneally (i.p.) injected with HP23-E6-IDL, YIK, and YIK-C16, respectively. The mouse serum samples were collected at different time point post-peptide treatment and tested for their inhibitory activity against HIV-1IIIB infection. As shown in Figure 4a, serum samples from mice treated with HP23-E6-IDL and YIK reached maximum inhibitory activity (about 300-fold of their IC_50_) at 1 h post-injection and decreased to the background level at about 2 h post-injection. In contrast, sera samples from mice treated with YIK-C16 showed inhibition peak (about 4500-fold of its IC_50_) at about 3 h post-injection and maintained high inhibitory activity (>275-fold of its IC_50_) for 15 h post-injection (Figure 4a). These results confirm that the addition of lipid domain C16 to the YIK peptide can significantly enhance its in vitro and ex vivo anti-HIV-1 activity.

Most recently, Chong et al. have reported that the concentration of an anti-HIV-1 lipopeptide in sera of rats treated with this lipopeptide is closely correlated with its ex vivo anti-HIV-1 activity [19]. Therefore, based on the in vitro IC_50_ and ex vivo anti-HIV-1 activity of peptides HP23-E6-IDL, YIK, and YIK-C16 for inhibiting HIV-1_IIIB_ infection described above, we estimated the concentration of the active peptides in sera of mice collected at different time points post-injection (Figure 4b). Based on the concentration profile, we calculated the serum half-life of HP23-E6-IDL, YIK, and YIK-C16, as well as other pharmacokinetic parameters of YIK-C16 (Table 2), using MODFIT software [20]. The serum half-life of YIK-C16 (t_1/2_ = 5.9 h) was about 4.5- and 5.9-fold longer than that of YIK (t_1/2_ = 1.3 h) and HP23-E6-IDL (t_1/2_ = 1.0 h), respectively. These results suggest that conjugation of lipid domain C16 to the YIK peptide can significantly extend the serum half-life of the peptide.

### 2.5. YIK-C16 Exhibited no In Vitro Cytotoxicity

To determine whether lipopeptide YIK-C16 has in vitro cytotoxicity, we incubated lymphocyte cell lines MT-2 and M7 with HP23-E6-IDL, YIK, or YIK-C16 at graded concentrations for three days and tested for their cell viability by CCK8 assay. As shown in Figure 5, none of these peptides exhibited in vitro cytotoxicity to MT-2 or M7 cells at concentrations as high as 8 μM, which is about 105- and 131-fold higher than the IC_50_ of YIK-C16 for inhibiting HIV-1IIIB and HIV-1Bal infection, respectively, suggesting that YIK-C16 has a good safety profile.

## 3. Discussion

To overcome the limitation of low potency and short in vivo half-life of the peptidic anti-HIV drug enfuvirtide (T20), we introduced a mutation, T639I, and added palmitic acid to HP23-E6-IDL, a peptide-based HIV fusion inhibitor with high potency and good resistance profile [11]. The newly conjugated peptide YIK-C16 is about 4- and 13-fold more potent than HP23-E6-IDL against HIV-1_IIIB_ and HIV-1_Bal_ infection, respectively, and about 16-fold more effective than HP23-E6-IDL against infection by HIV-1 mutants resistant to T20 and other HIV fusion inhibitory peptides, T2635 and HP23. Mechanistic study suggests that the improved anti-HIV-1 activity of lipopeptide YIK-C16 could result from the enhanced activity of binding to the target cell or viral membranes, not increased inhibitory activity on 6-HB formation.

It has been previously reported that lipids, such as cholesterol and sphingolipids, are assembled as lipid rafts on the cell membranes and play an essential role in viral entry and release [21,22,23,24,25]. Meanwhile, alteration in the ratio of these lipids on the viral membrane could significantly impair viral infectivity [26,27]. Even though the exact mechanism of action of T20 is still under debate, it has been reported that T20 inhibits HIV fusion by binding to the viral gp41 NHR domain via the N-terminal portion of T20 and then interacting with the target cell membrane via its C-terminal hydrophobic lipid-binding domain (LBD) [28,29,30]. Through its lipid domain, a lipopeptide can bind to cell membranes more efficiently than LBD of T20, thus possessing improved antiviral activity [12,13,31]. The highly potent anti-HIV-1 activity of YIK-C16 further certified the feasibility of this strategy and provided more insight into the functions of membrane protein and the roles of lipids in HIV-1 entry.

The inhibitory activity of YIK-C16 reached picomolar level. Even more promising, its serum half-life in mice is 4.5- and 5.9-fold longer than that of HP23-E6-IDL and YIK, respectively. Notably, C16 can bind to human serum albumin (HSA) [32,33], a human protein with wide distribution, with an in vivo half-life of 15 to 19 days [34]. Albuvirtide (ABT), a 3-maleimidopropionic acid (MPA)-modified HIV fusion inhibitor, is the only long-acting anti-HIV drug approved by the China Food and Drug Administration (CFDA) in 2018 for clinical use once a week because MPA can irreversibly bind to HSA, resulting in an extended half-life of the peptide (t_1/2_ = 25.8 h in rat) [35]. Since YIK-C16 may also bind to HSA via its C16 domain, its half-life may be further extended in humans. Meanwhile, YIK-C16 is about 10-fold more potent in inhibiting HIV-1 infection than ABT [36], making it a promising candidate for further development as a new long-acting anti-HIV drug for clinical use. Recently, stearic acid (C18)-modified anti-HIV peptide showed even higher potency and longer half-life time [19] than the C16-modified HIV fusion inhibitor. Therefore, it is possible that we could also modify YIK with C18 to further extend its in vivo half-life time.

Mutation T639I enhanced the antiviral activity of C34 in our previous study. However, it failed to improve the anti-HIV-1 activity of HP23-E6-IDL. This could be explained in two ways. First, many studies have revealed that only single-site mutation can elicit significant alteration of HIV-1 Env structure [37,38]. Conjugation of a tail anchor (IDL) to helical peptide (HP23-E6) may slightly alter the conformation of the original peptide, thus resulting in the mismatch of T639 to the original target residue on NHR. Therefore, mutation T639I had no effect on the antiviral activity of HP23-E6-IDL. Second, addition of hook-like structure to both N- and C-termini has already rendered peptide HP23-E6-IDL as potent as C34-T639I [15]. Thus, the potential effect of introducing T639I into HP23-E6-IDL may be ignored.

In sum, we have demonstrated that addition of a lipid domain, palmitic acid (C16), to YIK peptide, a potent HIV fusion inhibitor with two tail-anchors, results in significant enhancement of in vitro and ex vivo anti-HIV-1 activity and remarkably prolonged in vivo half-life, making it a good candidate for development into a new HIV fusion inhibitor-based, long-acting anti-HIV drug.

## 4. Materials and Methods

### 4.1. Peptides, Virus, and Cells

Peptides (Figure 1a) were synthesized by Synpeptide Co., Ltd. (Shanghai, China) with purity >95%. Peptides were dissolved in PBS, and their concentrations were determined by using NanodropTM 2000 spectrophotometers (Thermo Fisher Scientific Inc., Waltham, MA) and calculated based on a theoretical molar-extinction coefficient according to the peptide sequences as described previously [39].

HIV-1 IIIB, Bal and T20-resistant strains, MT-2 cells and H9/IIIB cells, as well as HIV-1 LAI and NL4-3 infectious molecular clones, were obtained from the National Institutes of Health (NIH) AIDS Reagent Program. CEMx174 5.25 M7 cells were kindly provided by Dr. C. Cheng-Mayer. T2635-resistant HIV-1 strains were produced by introducing plasmids provided by Dr. R.W. Sanders [40] into HIV-1 LAI infectious molecular clone. HP23-resistant HIV-1 NL4-3 infectious molecular clones were produced as previously described by Su et al. [41].

### 4.2. Inhibition Against HIV-1 Infection by Peptides

Inhibitory activities of peptides on infection by HIV-1 X4 strain IIIB, R5 strain Bal, and T20-, T2635-, HP23-resistant strains were determined as previously described [42]. For each well of a 96-well plate, 50 μL of a peptide were mixed with 50 μL of 100 × TCID50 (50% tissue culture infective doses) of HIV-1 live virus and incubated at 37 °C for 30 min. Afterwards, 2 × 10^4^ MT-2 (for X4 virus) or CEMx174 5.25 M7 cells (for R5 virus) were added. After overnight culture, the supernatant was replaced with fresh RPMI-1640 medium containing 10% fetal bovine serum (FBS). After further culture at 37 °C for three days, 50 μL of the culture medium were collected and mixed with equal volume of 5% (*v*/*v*) Triton X-100. The p24 antigen, which represents HIV-1 quantity, was detected by ELISA. Briefly, the collected mixtures were added to a plate coated with anti-HIV Immune Globulin (HIVIG) from the NIH AIDS Reagent Program. Anti-p24 mAb 183, rabbit anti-mouse IgG-HRP (Dako, Glostrup, Denmark) and substrate 3,3,5,5-TMB (Sigma-Aldrich, New York, NY) were added and washed away sequentially. The absorbance at 450 nm (A450) was determined by a Multi-Detection Microplate Reader (Ultra 384, Tecan, Tokyo, Japan). IC_50_s were calculated using Calcusyn software (Biosoft, Ferguson, MO), and the lines of best fit were drawn using GraphPad Prism 8 software (La Jolla, CA).

To determine whether a peptide binds with the target cells in order to execute its anti-HIV-1 activity, cell wash and nonwash assays were performed as previously described [43]. Briefly, 2 × 10^4^ MT-2 cells were incubated with a peptide at 4 °C for 30 min, followed by washes of the pretreated cells to remove the unbound peptide (control: without washes) before addition of HIV-1IIIB (100 × TCID50). After culture at 37 °C overnight, the supernatant was replaced by fresh RPMI-1640 medium containing 10% FBS. The inhibitory activity of the cell-bound peptide in the wash group and free peptide in the nonwash group was determined as described above.

To determine whether a peptide binds with the viral particle in order to execute its anti-HIV-1 activity, virus wash and nonwash assays were performed as previously described [43]. Briefly, HIV-1_IIIB_ (50 × TCID_50_) were incubated with a peptide at 4 °C for 30 min, followed by washes of the pretreated viral particles to remove the unbound peptide (control: without washes) before being added to 2 × 10^4^ MT-2 cells. After culture at 37 °C overnight, the supernatant was replaced by fresh RPMI-1640 medium containing 10% FBS. The inhibitory activity of the virus-bound peptide in the wash group and free peptide in the nonwash group was determined as described above.

### 4.3. Circular Dichroism (CD) Spectroscopy

The secondary structure of the single peptides HP23-IDL, YIK, YIK-C16 or N46, or the complexes HP23-IDL/N46, YIK/N46, or YIK-C16/N46, were assessed by CD spectroscopy as described previously [44]. Briefly, N46 and/or CHR-peptide in PBS (final concentration: 10 μM) was incubated at 37 °C for 30 min and then measured on a Jasco spectropolarimeter (Model J-815; Jasco, Inc., Easton, MD), using a 1-nm bandwidth with a 1-nm step resolution from 195 to 260 nm at room temperature. The baseline curve was determined on PBS alone.

### 4.4. HIV-1-Mediated Cell-Cell Fusion Assay

A dye transfer assay was performed to detect HIV-1 Env-mediated cell-cell fusion as described previously [42]. Briefly, for each well of a 96-well plate, 2 × 10^3^ H9/IIIB cells labeled with the fluorescent reagent Calcein AM (Molecular Probes, Inc., Eugene, Oregon) were incubated with a CHR-peptide at 37 °C for 30 min. Afterwards, 10^4^ MT-2 cells were added and incubated at 37 °C for 2 h. The fused and unfused Calcein-labeled H9/IIIB cells were counted under an inverted fluorescence microscope (Zeiss, Oberkochen, Germany). The IC_50_ values were calculated by using the Calcusyn computer program (Biosoft, Ferguson, MO).

### 4.5. Inhibition of 6-HB Formation by Peptides In Vitro

The inhibitory activity of a peptide on gp41 6-HB formation was detected as previously described [45]. Briefly, 50 μL of a peptide were incubated with 50 μl of N46 (4 μM) at 37 °C for 30 min. Then, 100 μL of C34 (2 μM) were added and incubated at 37 °C for another 30 min. Then, 50 μL of the mixture were added to a 96-well plate coated with 4 μg/mL polyclonal antibody against gp41 6-HB, NY364. Afterwards, a mAb specific for 6-HB, NC-1, rabbit anti-mouse IgG-HRP (Dako, Denmark) and substrate 3,3,5,5-TMB (Sigma-Aldrich, New York, NY) were added and washed away sequentially. The absorbance at 450 nm (A450) was determined by a Multi-Detection Microplate Reader. IC_50_s were calculated using Calcusyn software, and the lines of best fit were drawn using GraphPad Prism software.

### 4.6. Ex Vivo Anti-HIV-1 Activity and Serum Half-Life of Peptides

Evaluation of ex vivo anti-HIV-1 activity of a peptide in a mouse model was performed as described recently [46]. This mouse experiment was conducted under ethical guidelines and approved by the Institutional Laboratory Animal Care and Use Committee at Fudan University (20160927-2). ICR mice used for this study were bred at the Department of Laboratory Animal Science of Fudan University. Briefly, 5 mg/kg of a peptide were intraperitoneally (i.p.) injected into four female ICR mice (8 weeks). Mouse serum samples were collected before (0 h) and after injection (0.5, 1, 2, 4, and 6 h for peptides HP23-E6-IDL and YIK; 1, 3, 7, 11, 13, 15, 17, and 19 h for lipopeptide YIK-C16). Anti-HIV-1 activities of the mouse serum samples were determined in the same way as described above. The highest dilution-fold of the serum causing 50% inhibition of HIV-1_IIIB_ infection (IC_50_) was calculated. The concentration of an active peptide in serum of a mouse treated with this peptide was estimated based on the IC_50_ value of the peptide in the mouse serum for inhibiting HIV-1 infection activity as described above. Then, the serum half-life and other pharmacokinetic parameters of the peptide were calculated using MODFIT software [18] based on the estimated concentration of the active peptide in serum samples collected from the mouse at the different time points post-injection of the peptide.

### 4.7. Cytotoxicity of YIK-C16

The cytotoxic effects of HP23-E6-IDL, YIK, and YIK-C16 to MT-2 and M7 cells were determined as previously described [47]. A peptide at graded concentration was incubated with 2 × 10^5^/mL cells at 37 °C for 3 days before adding 10 μL of CCK8 reagent (Sigma-Aldrich). After another round of incubation at 37 °C for 2 h, A450 was measured with the Multi-Detection Microplate Reader. Cell viability was calculated by dividing A450 of untreated cells by A450 of cells treated with a peptide.

## Figures and Tables

**Figure 1 molecules-24-01134-f001:**
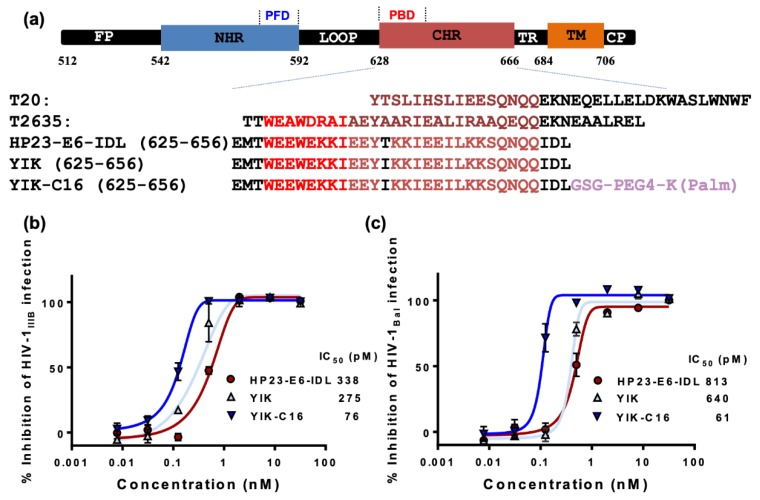
Sequence and anti-HIV-1 activity of the peptides. (**a**) Schematic diagram of HIV-1 gp41 functional domain and sequences of peptides derived from the gp41 CHR domains. FP, fusion peptide region; NHR and CHR, N- and C-terminal heptad repeats, respectively; PFD and PBD, pocket-forming and binding domains, respectively; TR and TM, tryptophan-rich and transmembrane regions, respectively. (**b**) Inhibitory activities of HP23-E6-IDL, YIK, and YIK-C16 against HIV-1_IIIB_ (X4 virus) infection; (**c**) Inhibitory activities of HP23-E6-IDL, YIK, and YIK-C16 against HIV-1_Bal_ (R5 virus) infection. Error bars in this figure represent standard deviations.

**Figure 2 molecules-24-01134-f002:**
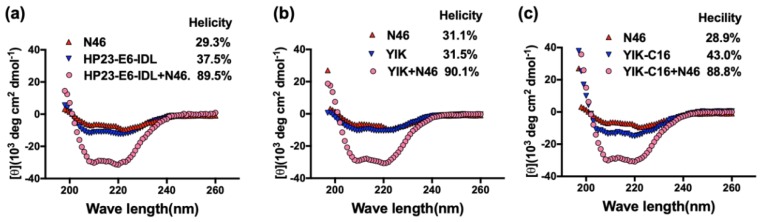
Secondary structure of HP23-E6-IDL, YIK, and YIK-C16, alone, or in complex with N46. The secondary structures of HP23-E6-IDL (**a**), YIK (**b**), and YIK-C16 (**c**), alone, or in complex with N46, were analyzed with circular dichroism (CD) spectroscopy. The CD spectra of HP23-E6-IDL/N46, YIK/N46, and YIK-C16/N46 complexes displayed typical double minima at 208 and 222 nm for the α-helical feature.

**Figure 3 molecules-24-01134-f003:**
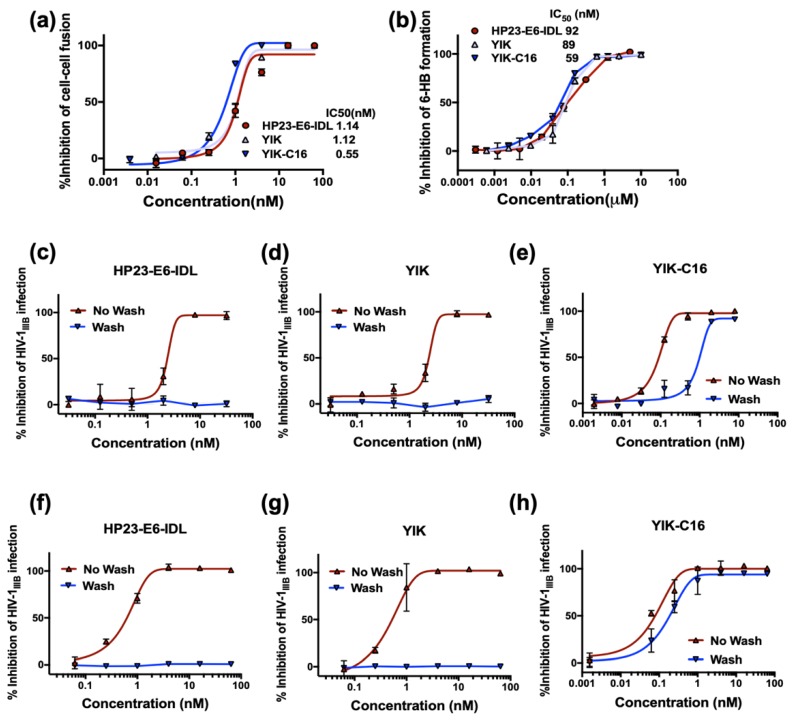
Inhibitory activity of YIK-C16. (**a**) Inhibitory activities of HP23-E6-IDL, YIK, and YIK-C16 against cell-cell fusion between H9/IIIB cells and MT-2 cells; (**b**) Inhibitory activities of HP23-E6-IDL, YIK, and YIK-C16 against 6-HB formation between N46 and C34 peptides; (**c**–**e**) Inhibitory activities of HP23-E6-IDL, YIK, and YIK-C16 against HIV-1_IIIB_ (X4 virus) infection when cells preincubated with peptides were washed to remove the unbound peptides, or not washed before addition of virus; (**f**–**h**) Inhibitory activities of HP23-E6-IDL, YIK, and YIK-C16 against HIV-1_IIIB_ (X4 virus) infection when virus preincubated with peptides were washed to remove the unbound peptides, or not washed before addition of target cells.

**Figure 4 molecules-24-01134-f004:**
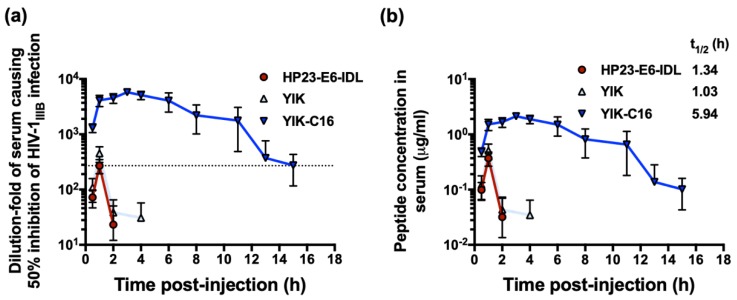
Ex vivo anti-HIV-1 activity and concentration of a peptide in serum samples of the peptide-treated mice. (**a**) Anti-HIV-1IIIB activity of the serum samples collected from mice at different time points after i.p. administration of HP23-E6-IDL, YIK, and YIK-C16. (**b**) Concentration of the active peptides in the serum samples was estimated.

**Figure 5 molecules-24-01134-f005:**
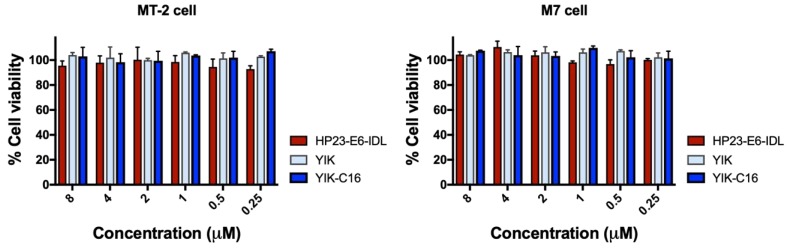
In vitro cytotoxicity of HP23-E6-IDL, YIK, and YIK-C16. The viability of MT-2 and M7 cells treated with HP23-E6-IDL, YIK, or YIK-C16 at graded concentrations was evaluated by CCK8 assay.

**Table 1 molecules-24-01134-t001:** Inhibitory activities of HP23-E6-IDL, YIK, and YIK-C16 against infection by HIV-1 mutants resistant to T20, T2635, or HP23.

Viruses	^a^ IC50 (pM)		
	HP23-E6-IDL	YIK	YIK-C16
T20-Resistant Strains			
HIV-1 NL4-3 D36G (WT)	912 ± 29	784 ± 32	69 ± 8
(D36G) V38A	647 ± 24	627 ± 25	99 ± 4
(D36G) V38A, N42D	1543 ± 253	1423 ± 184	179 ± 4
(D36G) V38E, N42S	748 ± 12	708 ± 50	68 ± 6
(D36G) V38A, N42T	838 ± 31	771 ± 21	40 ± 3
(D36G) N42T, N43K	941 ± 19	769 ± 22	60 ± 7
T2635-Resistant Strains			
HIV-1 LAI (WT)	932 ± 59	460 ± 18	83 ± 3
A6V	349 ± 19	773 ± 26	106 ± 7
Q66R	496 ± 168	707 ± 216	65 ± 10
K90E	1423 ± 242	885 ± 288	98 ± 6
K154Q	1246 ± 249	1490 ± 401	71 ± 7
Q79E/N126K	1312 ± 59	1150 ± 263	84 ± 8
K90E/N126K	842 ± 66	797 ± 22	81 ± 3
HP23-Resistant Strains			
HIV-1 NL4-3 (WT)	699 ± 25	856 ± 30	65 ± 8
E49K	2045 ± 126	1823 ± 72	115 ± 21
E49K/N126K	2563 ± 318	1378 ± 228	81 ± 13
D36G/E49K/N126K	3142 ± 400	2823 ± 611	100 ± 11
L34S/D36G/E49K/E136G	4937 ± 1100	5286 ± 569	188 ± 33

^a^ IC50 data were derived from the results of three independent experiments and expressed as means ± SD.

**Table 2 molecules-24-01134-t002:** Pharmacokinetic parameters of YIK-C16 in mice.

Parameter	Unit	Mice (i.p., *n* = 3)
Tmax	h	4.0 ± 1.7
Cmax	μg/mL	2.2 ± 0.1
t½	h	5.9 ± 3.2
AUC0-15h	h*μg/mL	14.9 ± 3.8
AUCINF_obs	h*μg/mL	15.0 ± 3.7
Vz_F_obs/Vz_obs	mL/kg	2738.8 ± 740.7
Cl_F_obs/Cl_obs	mL/h/kg	346.4 ± 76.1
MRTlast	h	9.5 ± 5.1

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
