# Peer review of "A Peptide-Based HIV-1 Fusion Inhibitor with Two Tail-Anchors and Palmitic Acid Exhibits Substantially Improved In Vitro and Ex Vivo Anti-HIV-1 Activity and Prolonged In Vivo Half-Life"

_molecules, 2019, doi:10.3390/molecules24061134_

Round 1

Reviewer 1 Report

The article describes an interesting work that tested two novel peptides that modify previously developed HP23-E6-IDL, a peptide with anti-HIV-1 activity. The novel forms introduced one amino acid substitution and a lipidic tail. The latter appears to increase the anti-HIV activity, without generating cytotoxicity.

The work appears well done and well presented. I have some concerns that can be considered minor points.

Lines 38-46: add references.

Fgure 1: quality is very low, amino acid sequences are difficult to understand and the T/I substitution is not readable.

Lines 78-83: The phrase appears difficult to understand, it should be simplified, or rephrased, or splitted.

Figure 3: The title appears not appropriate, because the experiments do not indicate the "mechanism" of action of YIK-C16, just its activity in inhibition of the different experiments (this is not sufficient to describe the "mechanism" of action).

Figure 4: try to present (a) (b) and (c) panels all together in a single panel, as for panel (d)

CD analysis is only qualitative, it should include the evaluation of helical content for each spectrum; alternatively, the method section should be corrected (in the present form of the manuscript, the secondary structure content does not appear "assessed").

Author Response

Reviewer 1

1.     Lines 38-46: add references.

Response: We have added 2 references and 1 website to provide proof for these sentences in the revised manuscript.

2.     Figure 1: quality is very low, amino acid sequences are difficult to understand and the T/I substitution is not readable.

Response: We have revised the Figure 1 and used larger font for peptide sequence to improve the figure's quality.

3.     Lines 78-83: The phrase appears difficult to understand, it should be simplified, or rephrased, or splitted.

Response: We thank the reviewer for this constructive suggestion. We changed this sentence into

“As shown in Figure 1b and c, the inhibitory activity of YIK against HIV-1 infection was only slightly better than that of HP23-E6-IDL. However, the resultant lipopeptide YIK-C16 possessed dramatically increased anti-HIV-1 activity in vitro. Its half maximum inhibitory concentrations (IC50) for inhibiting HIV-1IIIB and HIV-1Bal infection are 76 and 61 pM, respectively, which are about 4.4- and 13.3-fold more potent than HP23-E6-IDL.”.

4.     Figure 3: The title appears not appropriate, because the experiments do not indicate the "mechanism" of action of YIK-C16, just its activity in inhibition of the different experiments (this is not sufficient to describe the "mechanism" of action).

Response: We have changed the legend tittle of Figure 3 to “Inhibitory activity of YIK-C16”.

5.     Figure 4: try to present (a) (b) and (c) panels all together in a single panel, as for panel (d)

Response: We have integrated Figure 4a, b, and c into one figure and revised the figure legend accordingly.

6.     CD analysis is only qualitative, it should include the evaluation of helical content for each spectrum; alternatively, the method section should be corrected (in the present form of the manuscript, the secondary structure content does not appear "assessed").

Response: We have supplemented the estimated helix content to Figure 2.

Reviewer 2 Report

In the manuscript entitled “A peptide-based HIV-1 fusion inhibitor with two tail-anchors and palmitic acid exhibits substantially improved in vitro and ex vivo anti-HIV-1 activity and prolonged in vivo half-life”, Su et al. have designed, synthesized, and evaluated a lipid-conjugated peptide, YIK-C16. They found that YIK-C16 is at least 10-fold more potent than non-lipid-conjugated peptides in inhibiting infection of divergent HIV-1 strains tested, including those resistant to enfuvirtide (T20), the first US FDA-approved anti-HIV peptide drug. More importantly, YIK-C16 showed much longer in vivo half-life than the non-lipid-conjugated peptides. The serum of YIK-C16-treated mice could effectively inhibit HIV-1 infection at 15 h post-administration of peptide, suggesting that YIK-C16 has a good potential to be further developed as a long-acting anti-HIV drug.

This manuscript is well-written and suitable for publication in Molecules after minor revision as described below:

1.       The authors have demonstrated that the increased antiviral activity of YIK-C16 is attributed to its binding to the target cell membrane as shown in a wash assay, i.e., cells pretreated with YIK-C16 were washed to remove unbound lipid peptide before addition of virus. Most recently, Si L et al (Sci Adv. 2018; 4:eaau8408) also performed a wash assay, but in a different way, to study the mechanism of an antiviral agent. The virus was pretreated with the antiviral agent, followed by washing the virus to remove the unbound antiviral agent, before addition of the cell. I suggest the authors to also perform this experiment, in order to determine whether YIK-C16 can also act on the virus membrane.

2.       Please provide more detail information about the "Wild-type" HIV-1 strains in Table 1.

3.       The dash lines in Fig. 2 and Fig. 3b should be removed.

4.       The "μ" in the X-axis [Concentration (μM)] of Fig. 3b, Fig. 5a and 5b is unclear. It should be changed.

Author Response

Reviewer 2

1.     The authors have demonstrated that the increased antiviral activity of YIK-C16 is attributed to its binding to the target cell membrane as shown in a wash assay, i.e., cells pretreated with YIK-C16 were washed to remove unbound lipid peptide before addition of virus. Most recently, Si L et al (Sci Adv. 2018; 4:eaau8408) also performed a wash assay, but in a different way, to study the mechanism of an antiviral agent. The virus was pretreated with the antiviral agent, followed by washing the virus to remove the unbound antiviral agent, before addition of the cell. I suggest the authors to also perform this experiment, in order to determine whether YIK-C16 can also act on the virus membrane.

Response: We thank the reviewer for this constructive suggestion. We performed this experiment as described by Si, L. et al (reference 43). We found that when virus particles pretreated with YIK-C16 were washed to remove the unbound peptide, HIV-1IIIB infection could till be inhibited, while those pre-treated by HP23-E6-IDL and YIK maintained their infectivity after the unbound peptides were removed by washes, suggesting that YIK-C16 can also bind to HIV-1 membrane to exert its antiviral activity.

We have added the result in the revised manuscript:

“We then tested whether YIK-C16 could also bind to viral membrane to inhibit HIV-1 infection. HIV-1IIIB viral particles were incubated with peptides at 4°C for 30 min and then washed to remove the unbound peptides, or remained unwashed to serve as a control, and were then applied to infect MT-2 cells. As shown in Figure 3f and 3g, HP23-E6-IDL and YIK could inhibit HIV-1IIIB infection without wash-away of peptides whereas after washing, neither HP23-E6-IDL nor YIK could inhibit HIV-1IIIB infection. In contrast, YIK-C16 could inhibit HIV-1IIIB infection when the unbound peptide, suggesting that YIK-C16 could also bind to the viral membrane to enhance its antiviral activity.”

2.     Please provide more detail information about the "Wild-type" HIV-1 strains in Table 1.

Response: We have supplemented more detail information about the "Wild-type" HIV-1 strains in Table 1.

3.     The dash lines in Fig. 2 and Fig. 3b should be removed.

Response: We have removed the dash line in Fig. 2 and Fig. 3b.

4.     The "μ" in the X-axis [Concentration (μM)] of Fig. 3b, Fig. 5a and 5b is unclear. It should be changed

Response: We have modified Fig. 3b, Fig. 5a and 5b accordingly.